# Greater improvement in aerobic capacity after a polarized training program including cycling interval training at low cadence (50–70 RPM) than freely chosen cadence (above 80 RPM)

**Rafal Hebisz, Paulina Hebisz** \*

Department of Physiology and Biochemistry, Wroclaw University of Health and Sport Sciences, Wroclaw, Poland

\* paulina.hebisz@awf.wroc.pl

## Abstract

This study compared the impact of two polarized training programs (POL) on aerobic capacity in well-trained (based on maximal oxygen uptake and training experience) female cyclists. Each 8-week POL program consisted of sprint interval training (SIT) consisting of 8–12 repetitions, each lasting 30 seconds at maximal intensity, high-intensity interval training (HIIT) consisting of 4–6 repetitions, each lasting 4 minutes at an intensity of 90–100% maximal aerobic power, and low-intensity endurance training (LIT) lasting 150–180 minutes with intensity at the first ventilatory threshold. Training sessions were organized into 4-day microcycles (1st day—SIT, 2nd day—HIIT, 3rd day—LIT, and 4th day—active rest), that were repeated throughout the experiment. In the first POL program, exercise repetitions during SIT and HIIT training were performed with freely chosen cadence above 80 RPM ($POL_{FC}$ group, n = 12), while in the second POL program with low cadence 50–70 RPM ($POL_{LC}$ group, n = 12). Immediately before and after the 8-week POL intervention, participants performed an incremental test to measure maximal aerobic power (Pmax), power achieved at the second ventilatory threshold (VT2), maximal oxygen uptake ($VO_2$max), maximal pulmonary ventilation (VEmax), and gross efficiency (GE). Moreover, participants performed $VO_2$max verification test. Analysis of variance showed a repeated measures effect for Pmax (F = 21.62; $\eta^2$ = 0.5; p = 0.00), $VO_2$max (F = 39.39; $\eta^2$ = 0.64; p = 0.00) and VEmax (F = 5.99; $\eta^2$ = 0.21; p = 0.02). A repeated measures x group mixed effect was demonstrated for Pmax (F = 4.99; $\eta^2$ = 0.18; p = 0.03) and $VO_2$max (F = 6.67; $\eta^2$ = 0.23; p = 0.02). Post-hoc Scheffe analysis showed that increase in Pmax were statistically significant only in $POL_{LC}$ group. The Friedman test showed that VT2 differed between repeated measures only in the $POL_{LC}$ group ($\chi^2$ = 11; p = 0.001; W = 0.917). In conclusion, it was found that POL program where SIT and HIIT were performed at low cadence was more effective in improving aerobic capacity in well-trained female cyclists, than POL with SIT and HIIT performed at freely chosen cadence. This finding is a practical application for athletes and coaches in cycling, to consider not only the intensity and duration but also the cadence used during various interval training sessions.

**Data Availability Statement:** The data supporting this study's findings are available at: doi.org/10. 18150/O77ZL8.

**Funding:** The author(s) received no specific funding for this work.

**Competing interests:** The authors have declared that no competing interests exist.

# Introduction

Polarized training program (POL) is effective in improving cyclists' aerobic capacity [1–3]. POL is described as a training cycle characterized by polarization of training intensity and incorporates low-intensity training as well as high-intensity training [2,4]. The volume of low-intensity training sessions is approximately 80% of the total training volume, while the high-intensity training is approximately 20% of the total training volume [2,4,5]. In polarized training programs, moderate-intensity training at the level of the lactate threshold or the second ventilatory threshold (VT2) is not used, or these training sessions account for a small part of the training program (approximately 5% of the total training volume) [6,7]. Hebisz et al. [8,9] showed that aerobic capacity assessed by maximal oxygen uptake ($VO_2max$) increased by 12– 14% in well-trained cyclists as a result of POL. In the studies Hebisz et al. [8,9] applied POL in which training microcycles consisted of sprint interval training (SIT) included 8 to 20 repetitions of 30-s maximal intensity cycling, high-intensity interval training (HIIT) included 4 to 7 repetitions of 4 min cycling at an intensity of 90–100% maximal aerobic power, and low-intensity endurance training (LIT) involved 120–180 min of cycling at an intensity of the first ventilatory threshold.

The concept of a polarized training program is also used in other sports disciplines. Filipas et al. [10] showed that POL is effective in improving aerobic capacity in well-trained runners, as it increased $VO_2max$, power at lactate threshold and 5-km running time trial performance. Pla et al. [11] conducted a study among elite junior swimmers and showed that polarized training program elicited greater improvement than threshold training on 100-m time-trial performance, with less fatigue and better quality of recovery. In the triathletes group, POL increased power at ventilatory thresholds, maximal aerobic power (Pmax) and $VO_2max$ [12]. Similarly, in rowers, $VO_2max$ and 6-min ergometer rowing performance increased were observed as a result of POL [13]. In winter sports, the beneficial effect of POL has also been demonstrated. Speed skaters' performance improved and their lactate after competition decreased considerably after a polarized training program, compared with a threshold training program [14]. The polarized training program used in cross-country skiers affected $VO_2max$ increased, treadmill exercise time increased, and the recovery time decreased [15]. Interestingly, Kim et al. [15] showed that POL had a better effect on cardiorespiratory function in male than in female cross-country skiers. Therefore, when applying a polarized training program to athletes, it should be planned in detail by sex, exercise amount, intensity, and type of training [15]. The above cited studies indicate a beneficial effect of POL in various groups of athletes. This is confirmed by the systematic review with meta-analysis Oliveira et al. [16] indicating that polarized training intensity distribution was superior to other training intensity distribution regimens for endurance performance improvement. Particularly in shorter duration interventions (no longer than 12 weeks) and highly trained athletes. However, the effect of POL was similar to that of other programs among endurance athletes with lower performance and lower training levels [16].

The literature provides information that, in addition to polarizing training intensity, it is important to add strength training to improve aerobic capacity among endurance athletes [17–20]. It has been shown that concurrent strength and endurance training improves cycling performance, fractional utilization of $VO_2max$, cycling economy, and an increase in the proportion of type IIA muscle fibers at the expense of type IIX muscle fibers, both in women and men [17–19]. Some studies have reported that type IIA fibers are more economical than type IIX fibers [17,19]. Moreover, Rønnestad [20] reported that the inclusion of heavy strength training in the training program of elite cyclists improved their performance during sprints, and also their strength felt throughout the entire cycling race. Experimental studies examining

strength training focuses on the use of exercises with additional external load, such as the barbell half-squats or half-squats in a Smith machine [18,20]. However, in cycling training, it is possible to increase resistance and force a lower cadence by adjusting the gears on the bike and by selecting appropriate external conditions (e.g., performing an uphill cycling effort). Paton et al. [21,22] showed that systematically performed interval training with low cadence increased cyclists' physical performance. However, in this study, participants in the control group continued their training program without any changes [21,22]. There was no comparison group in which interval training would be performed at high or freely chosen cadence [21,22]. The literature lacks information on the effects of performing cycling training with a low cadence in highly trained female athletes, as well as information on the effects of simultaneous use of low cadence during SIT and HIIT cycling training, therefore, this became the aim of the presented study.

The presented study aimed to compare the changes in aerobic capacity as a result of two POL programs, consisting of SIT, HIIT, and LIT training, performed by well-trained female cyclists. In the first group, exercise repetitions during SIT and HIIT were performed with a freely chosen cadence, while in the second group, exercise repetitions during SIT and HIIT were performed with a low cadence. It was hypothesized that performing SIT and HIIT training with a low cadence would result in greater improvements in aerobic capacity, gross efficiency and maximal aerobic power, than performing these training with a freely chosen cadence.

## Materials and methods

### Study participants

The presented study involved 26 female, well-trained cyclists aged 17–20. The participants of the experiment were considered as well-trained based on their baseline $VO_2max$ and training experience, this classification was presented by Decroix et al. [23]. The participants were divided into two groups, formed using the matched pairs randomization method [24], after the participants had been assigned ranks according to the value of $VO_2max$. During the study, 2 participants (one from each group) were excluded due to infections. All participants performed the POL, consisting of SIT, HIIT, and LIT training. The first group (n = 12) performed exercise repetitions during SIT and HIIT training with a freely chosen cadence above 80 RPM–$POL_{FC}$ group. The second group (n = 12) performed exercise repetitions during SIT and HIIT training with a low cadence of 50–70 RPM–$POL_{LC}$ group. The characteristics of the study groups are shown in Table 1.

Each of the participants had at least 3 years of cycling training experience, trained at least 10 hours per week (excluding rest periods), and participated in at least 15 cycling races per year. During the three months preceding the experiment, all cyclists used the same training program included LIT training which constituted 90% of the total training volume, the

Table 1. Anthropometric and physiological characteristics of the studied groups.

|  | Body mass (kg) | Body height (m) | Age (years) | $VO_2max$ (ml·min$^{-1}$·kg$^{-1}$) | Pmax (W) |
|---|---|---|---|---|---|
| $POL_{FC}$ | 55.5±5.4 | 1.65±0.08 | 17.9±1.3 | 54.4±5.4 | 259.8±44.4 |
| $POL_{LC}$ | 56.6±4.7 | 1.67±0.05 | 18.0±0.7 | 54.5±4.7 | 268.8±30.3 |

$VO_2max$–maximal oxygen uptake; Pmax–maximal aerobic power; $POL_{FC}$–the group performing POL program in which exercise repetitions during SIT and HIIT training were performed with freely chosen cadence above 80 RPM; $POL_{LC}$–the group performing POL program in which exercise repetitions during SIT and HIIT training were performed with a low cadence of 50–70 RPM.

remaining 10% consisted of: core stability training [25], running with jumping exercises–similar to de Poli et al. [26], and lower limb strength training–similar to Vikmonen et al. [18].

The study design was approved by the Ethics Committee of the Wroclaw University of Health and Sport Sciences (protocol code: 39/2019; date of approval: 26 November 2019) and implemented following the Declaration of Helsinki. The course of the study, the study procedures, and the potential benefits and risks of participation were explained in detail to the cyclists and their legal guardians. Written informed consent to participate in the study was obtained from the cyclists and their guardians. Each participant was also informed that they could withdraw from the experiment at any time without giving a reason. Before starting the experiment, it was checked whether each participant had consent to practice cycling from a sports medicine physician. In addition, blood pressure was measured using an aneroid sphygmomanometer (Riester, Germany) in a sitting position before the experiment to exclude participants with hypertension from the experiment. Resting heart rate was recorded using Polar V800 monitors (Polar Electro Oy, Finland) throughout the experiment to exclude participants with tachycardia from the experiment. Each participant was informed that in the event of feeling unwell, dizziness, or shortness of breath, training should be discontinued. During the experiment, none of the participants experienced the above-described symptoms.

## Test procedures

Immediately before and after the 8-week POL intervention, participants performed an incremental test and the next day a test verifying $VO_2max$. Each of the tests was performed in controlled laboratory conditions (temperature 20˚C and humidity 45–50%), on a Lode Excalibur Sport cycle-ergometer (Lode BV, Groningen, the Netherlands). The period in which laboratory exercise tests were performed and the training intervention was applied lasted from 1/02/2021 to 30/04/2021.

**Incremental test.**   An incremental test consisted of several 3-minute steps. During each step, the effort was performed with constant intensity—power. When 3 minutes elapsed, the power was increased. The test began with a load of 40 W, and every 3 minutes the load was increased by 40 W until the participant refused to continue. To determine maximal aerobic power (Pmax) 0.22 W was subtracted from the final power for each missing second if the participant was unable to exercise for the entire 3 minutes at the last step [5,8,27].

During the test and during the 5-minute post-test recovery, respiratory parameters were recorded. The participants wore a mask connected to a Quark respiratory gas analyzer (Cosmed, Rome, Italy), which was calibrated before the test. Respiratory parameters, including oxygen uptake ($VO_2$), carbon dioxide excretion ($VCO_2$), and pulmonary ventilation (VE) were measured in each recorded breath (breath-by-breath) and then averaged at 30-second intervals. Peak oxygen uptake ($VO_2peak1$), peak pulmonary ventilation (VEpeak1), peak exercise respiratory exchange ratio (RER-ex), and peak recovery respiratory exchange ratio (RER-rec) were analyzed from the recorded data. Power at the first ventilatory threshold (VT1) was indicated at the point of the first nonlinear increase in $VE·VO_2^{-1}$ equivalent, power at the second ventilatory threshold (VT2) was indicated at the point of the second nonlinear increase in $VE·VO_2^{-1}$ equivalent and the increase in $VE·VCO_2^{-1}$ [28]. Two individuals independently indicated VT1 and VT2, if their indications were different, then the indication was made by a third person. Gross efficiency (GE) was calculated for the third minute of exercise at 120W, following the recommendations for determining GE for steady-state exercises at which RER < 1 [29]. GE was calculated based on the formula [30]: $GE = W · EE^{-1}$, where: W—work done, calculated based on the power achieved and exercise duration; EE—energy expenditure

calculated by the Quark respiratory gas analyzer software using the indirect calorimetry method.

**VO$_2$max verification test.**   The VO$_2$max verification test was performed according to the methodology of Hebisz et al. [31]. The test was preceded by a 15-minute warm-up consisting of 5 min exercise with the power achieved at VT1, then 10 min at a power between the VT1 and the VT2. The warm-up was followed by a 5-minute passive break. The test lasted 3 min and was performed with load of 110% Pmax (determined during incremental test). During the test, respiratory parameters were recorded using the Quark respiratory gas analyzer, similarly to incremental test. The values averaged every 30 s were used in data analysis. The highest recorded oxygen uptake (from the averaging of 30-s intervals) was taken as the peak oxygen uptake (VO$_2$peak2). The higher value between VO$_2$peak1 and VO$_2$peak2 was considered as the VO$_2$max. Using the same principle, maximal values for minute pulmonary ventilation (VEmax) was determined.

Differences for selected parameters: Pmax-diff, GE-diff, VO$_2$max-diff, VT1-diff, VT2-diff, and VEmax-diff, were calculated between measurements performed after and before the 8-week POL intervention.

## Training interventions

The training intervention lasted 8 weeks, all participants performed a polarized training program. Training sessions were performed outdoors, on participants' bicycles. During training sessions, power output was monitored using the PowerTap G3 ANT+ and GS ANT+ system (PowerTap, Madison, US) and heart rate was monitored using the Garmin Edge 520 and Edge 810 system (Garmin Ltd., Olathe, US), which were validated [32,33]. The POL intervention included sprint interval training (SIT), high-intensity interval training (HIIT), low-intensity endurance training (LIT) and active rest (AR). On the 1st day of experiment SIT was performed, on the 2nd day HIIT was performed, on the 3rd day LIT was performed, on the 4th day was AR, and again on the 5th day was SIT, on the 6th day was HIIT, on the 7th day was LIT and on the 8th day was AR, etc. throughout the experiment (4-day microcycle was repeated) (Fig 1). Characteristics of training sessions:

- Sprint interval training (SIT), in which 8–12 repetitions with maximal intensity were performed, each repetition lasting 30 seconds. The training was divided into sets. In each set 4 repetitions were performed, between repetitions was 90 second of active rest at low intensity–similar to the power at VT1.. Between sets was 25 minutes of active rest at also at low intensity. In the 1st-4th week of the training intervention, the cyclists performed 8 repetitions and the entire training session including warm-up lasted 80 minutes. In the 5th-8th week the cyclists performed 12 repetitions and the entire training session including warm-up lasted 110 minutes.. The POL$_{FC}$ group applied a freely chosen cadence above 80 RPM while the POL$_{LC}$ group applied a low cadence 50–60 RPM when performing repetitions during SIT. In order to obtain the appropriate power and cadence during SIT training, cyclists adjusted the bike gears and performed repetitions uphill with a slope 6–9%.

- High-intensity interval training (HIIT) consisted of 4–6 repetitions, each repetition lasted 4 minutes and was performed at an intensity of 90–100% Pmax. An 8-minute active rest was used between repetitions, at an intensity similar to the power achieved at VT1. In the 1st-4th week of the training intervention, the cyclists performed 4 repetitions and the entire training session including warm-up lasted 85 minutes. In the 5th-8th week the cyclists performed 6 repetitions and the entire training session including warm-up lasted 110 minutes. The POL$_{FC}$ group applied a freely chosen cadence above 80 RPM while The POL$_{LC}$ group applied

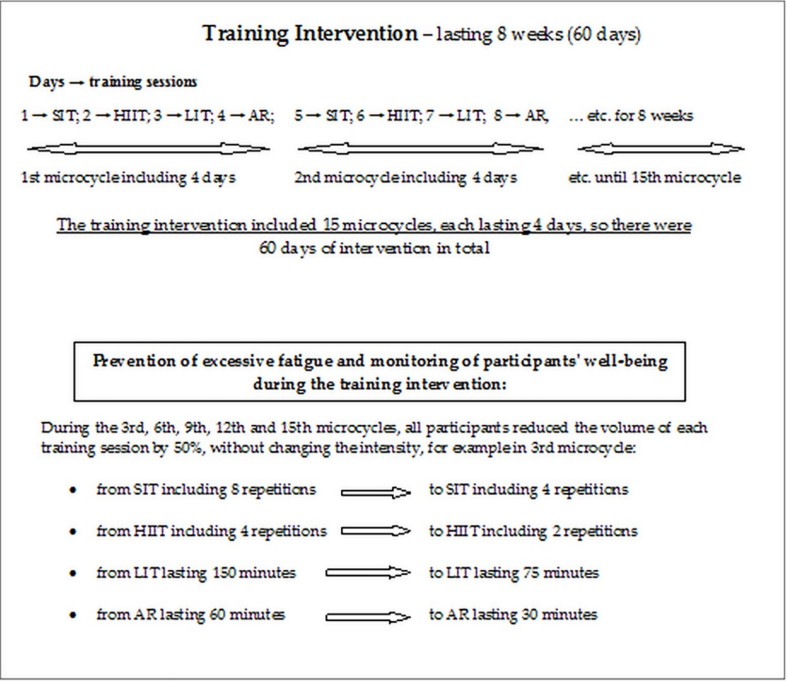

**Fig 1. Diagram showing the training intervention.** SIT—sprint interval training, HIIT—high-intensity interval training, LIT—low-intensity endurance training, AR—active rest.

a low cadence 60–70 RPM when performing repetitions during HIIT. Similar to SIT, to obtain the appropriate power and cadence, cyclists adjusted the bike gears and performed repetitions uphill with a slope 6–9%.

- Low-intensity endurance training (LIT), performed at an intensity close to the power achieved at VT1, lasting 150–180 minutes. In the 1st-4th week of the training intervention, the entire training session lasted about 150 minutes, and in the 5th-8th week– 180 minutes.

- Active rest (AR)– 60–80 minutes of exercise at 20% less power than at VT1. In the 1st-4th week of the training intervention, the active rest lasted about 60 minutes, and in the 5th-8th week– 80 minutes.

**Preventing overreaching and monitoring participants' well-being.** During the training intervention to prevent overreaching, all participants used recovery microcycles in which they reduced the volume of the main part of the training session by 50%, similar to the assumptions of undulating periodization of training [34], as shown in figure (Fig 1). In addition, during the training intervention, to prevent overload, daily monitoring of heart rate variability (HRV) was recorded using Polar V800 heart rate monitors (Polar Electro Oy, Kempele, Finland). The measurement was performed every day immediately after waking up, in a supine position. Data calculations were performed using Kubios HRV Standard software (Kubios Oy, Kuopio, Finladia). The mean artifact correction threshold was used when performing the analysis. The root mean square of successive differences between normal heartbeats (RMSSD) values were calculated for the part of the recording between the 30th and 150th seconds. The moving average for RMSSD (avRMSSD) over the past 7 days was then calculated. In HIIT, the training

load was corrected when the RMSSD measured before that training session was lower than avRMSSD by a value greater than the smallest worthwhile change (SWC). SWC was calculated as the mean ± 0.5SD. If a low RMSSD value was observed, training session of the same duration as planned was performed, but the training intensity was with power at VT1 (similar to the procedure described by Vesterinen et al. [35] and Javaloyes et al. [36]).

## Statistical analysis

Statistica 13.1 software was used for statistical calculations. Using the Shapiro-Wilk test, the distribution of the studied parameters was checked. Only in the $POL_{FC}$ group did the distribution of the RER-rec parameter differ from the normal distribution in the test performed after the POL intervention. Levene's test was used to analyze the homogeneity of variance among the analyzed parameters. It was shown that only VT2 does not meet the requirement of variance homogeneity. Therefore, in the further analysis of RER-rec and VT2, the Friedman test was used. The distribution of the other parameters did not differ from the normal distribution, so parametric tests were used in further analysis. To estimate the effects for the $POL_{FC}$ and $POL_{LC}$ groups, analysis of variance with repeated measures was used, followed by the Scheffe post-hoc test. To calculate the minimum required number of participants for statistically significant results, G-Power software was used. It was assumed that the aim was to obtain large statistical effects ($\eta^2$ above 0.14). Thus, it was determined that the minimum study group should be at least 24 participants. Using Pearson correlation, the strength of the correlation between Pmax-diff and the following parameters was calculated: GE-diff, $VO_2$max-diff, VT1-diff, VT2-diff, VEmax-diff. For parameters whose correlations were statistically significant, a multiple regression formula was calculated.

A statistical significance threshold of $p < 0.05$ was adopted for the analyses performed.

## Results

Using the analysis of variance, the effect of repeated measurements was demonstrated for Pmax ($F = 21.62$; $\eta^2 = 0.50$; $p = 0.00$), $VO_2$max expressed in $ml \cdot kg^{-1} \cdot min^{-1}$ ($F = 39.39$; $\eta^2 = 0.64$; $p = 0.00$), $VO_2$max expressed in $l \cdot min^{-1}$ ($F = 42.57$; $\eta^2 = 0.66$; $p = 0.00$), VT1 ($F = 19.23$; $\eta^2 = 0.47$; $p = 0.00$) and VEmax ($F = 5.99$; $\eta^2 = 0.21$; $p = 0.02$). A repeated measures x group mixed effect was also found for Pmax ($F = 4.99$; $\eta^2 = 0.18$; $p = 0.03$), $VO_2$max expressed in $ml \cdot kg^{-1} \cdot min^{-1}$ ($F = 6.67$; $\eta^2 = 0.23$; $p = 0.02$) and VEmax ($F = 4.58$; $\eta^2 = 0.17$; $p = 0.04$). The level of statistical significance for the post-hoc test is presented in Table 2. Freadmann's analysis of variance showed that VT2 differed between repeated measures only in the $POL_{LC}$ group ($\chi^2 = 11$; $p = 0.001$; $W = 0.917$).

It was shown that Pmax-diff correlated statistically significantly with $VO_2$max-diff ($r = 0.51$; $p < 0.05$), VT2-diff ($r = 0.60$; $p < 0.05$) and VEmax-diff ($r = 0.71$; $p < 0.05$). In the case of VT1-diff ($r = 0.29$; $p > 0.05$), and GE-diff ($r = 0.35$; $p > 0.05$), there were no statistically significant correlations with Pmax-diff (Fig 2). Multiple correlation was demonstrated, which is expressed in the following regression equation ($r = 0.81$; $F = 20,43$; $p = 0.000$): Pmax-diff = 4.69 + 0.26 · VT2-diff + 1.03 · VEmax-diff.

## Discussion

The presented study showed that the POL intervention using low cadence during exercise repetitions in SIT and HIIT training was more effective in improving aerobic capacity of well-trained female cyclists, than the POL using freely chosen cadence during SIT and HIIT. In the group of cyclists performing interval training with low cadence, Pmax, VT1, VT2, $VO_2$max (both expressed in $ml \cdot min^{-1} \cdot kg^{-1}$ and $l \cdot min^{-1}$) and VEmax increased. Whereas, in the group of

**Table 2. Changes in power, gross efficiency, and respiratory parameters after the POL interventions with a freely chosen cadence and a low cadence.**

| | Before POL intervention | | | After POL intervention | | | p |
|---|---|---|---|---|---|---|---|
| | mean ± SD | 95% CI | | mean ± SD | 95% CI | | |
| | | Lower | Upper | | Lower | Upper | |
| **$POL_{FC}$ group** | | | | | | | |
| **BM [kg]** | 55.5 ± 5.4 | 52.1 | 59.0 | 56.2 ± 5.5 | 52.7 | 59.7 | 0.394 |
| **Pmax [W]** | 259.8 ± 44.4 | 231.6 | 288.1 | 267.5 ± 43.6 | 239.8 | 295.2 | 0.423 |
| **GE [%]** | 18.0 ± 1.3 | 17.2 | 18.9 | 18.2 ± 1.0 | 17.5 | 18.8 | 0.970 |
| **$VO_2$max [ml·min$^{-1}$·kg$^{-1}$]** | 54.4 ± 5.4 | 50.9 | 57.8 | 56.4 ± 5.3 | 53.0 | 59.7 | 0.108 |
| **$VO_2$max [l·min$^{-1}$]** | 3.02 ± 0.42 | 2.75 | 3.28 | 3.16 ± 0.41 | 2.90 | 3.42 | **0.032** |
| **VT1 [W]** | 128.4 ± 22.9 | 113.9 | 143.0 | 138.5 ± 22.8 | 125.1 | 151.9 | 0.459 |
| **VT2 [W]** | 195.7 ± 39.8 | 170.5 | 221.0 | 203.0 ± 42.6 | 176.8 | 230.9 | 0.132 |
| **VEmax [l·min$^{-1}$]** | 121.0 ± 19.5 | 108.7 | 133.4 | 121.6 ± 18.9 | 109.6 | 133.6 | 0.997 |
| **RER-ex** | 1.13 ± 0.08 | 1.08 | 1.18 | 1.10 ± 0.05 | 1.07 | 1.13 | 0.345 |
| **RER-rec** | 1.53 ± 0.19 | 1.41 | 1.65 | 1.51 ± 0.17 | 1.40 | 1.62 | 0.366 |
| **$POL_{LC}$ group** | | | | | | | |
| **BM [kg]** | 56.6 ± 4.7 | 53.6 | 59.6 | 56.6 ± 4.6 | 53.7 | 59.5 | 0.999 |
| **Pmax [W]** | 268.8 ± 30.3 | 249.6 | 288.1 | 290.7 ± 29.0 | 272.2 | 309.1 | **0.001** |
| **GE [%]** | 18.3 ± 1.4 | 17.4 | 19.2 | 18.8 ± 0.9 | 18.2 | 19.3 | 0.487 |
| **$VO_2$max [ml·min$^{-1}$·kg$^{-1}$]** | 54.5 ± 4.7 | 51.5 | 57.5 | 59.3 ± 4.1 | 56.7 | 61.9 | **0.000** |
| **$VO_2$max [l·min$^{-1}$]** | 3.09 ± 0.43 | 2.81 | 3.36 | 3.36 ± 0.39 | 3.11 | 3.61 | **0.000** |
| **VT1 [W]** | 129.0 ± 14.8 | 119.6 | 138.4 | 157.1 ± 22.6 | 142.7 | 171.4 | **0.002** |
| **VT2 [W]** | 197.7 ± 28.8 | 179.4 | 215.9 | 232.3 ± 23.0 | 217.7 | 247.0 | **0.001** |
| **VEmax [l·min$^{-1}$]** | 120.4 ± 12.1 | 112.7 | 128.1 | 128.6 ± 13.7 | 121.0 | 136.3 | **0.032** |
| **RER-ex** | 1.11 ± 0.05 | 1.08 | 1.14 | 1.10 ± 0.04 | 1.07 | 1.13 | 0.957 |
| **RER-rec** | 1.47 ± 0.08 | 1.42 | 1.53 | 1.47 ± 0.15 | 1.38 | 1.57 | 0.564 |

POL–polarized training program; $POL_{FC}$ group–the cyclists performed exercise repetitions in SIT and HIIT training with a freely chosen cadence above 80RPM, during POL intervention; $POL_{LC}$ group–the cyclists performed exercise repetitions in SIT and HIIT training with a low cadence of 50-70RPM, during POL intervention; BM—participants' body mass; Pmax–maximal aerobic power; GE–gross efficiency; $VO_2$max–maximal oxygen uptake; VT1 –power achieved at the first ventilatory threshold; VT2 –power achieved at the second ventilatory threshold; VEmax–maximal minute pulmonary ventilation; RER-ex–peak exercise respiratory exchange ratio; RER-rec–peak recovery respiratory exchange ratio; mean–arithmetic mean value; SD–standard deviation value; CI: Lower and upper confidence intervals; p–the level of statistical significance for the post-hoc test or for the Friedman test (VT2, RER-rec).

cyclists performing interval training with freely chosen cadence, only $VO_2$max (expressed in l·min$^{-1}$) increased. When cyclists perform exercise with a cadence lower than the preferred (freely chosen) cadence, resistance increases [37]. If high-resistance training coexists with endurance training, better effects are observed in improving of cardiovascular fitness and athletic performance [18]. This effect is attributed to interference between the effects of resistance and endurance training in the muscles [38]. Del Vecchio et al. [39] showed that after 4 weeks of resistance training (weightlifting), the exercise recruitment of motor units increased. Perhaps similar adaptive changes were provoked in the presented study by the use of maximal- and high-intensity cycling efforts performed uphill and using heavy bicycle gears, which resulted in low cadence. Mikkola et al. [40], Paavolainen et al. [41] and Hyttinen and Häkkinen [42] indicated that a specific type of strength or resistance training in relation to a sports discipline and explosive-strength training may induce specific neural adaptations, such as an increase in the activation rate of motor units, while muscle hypertrophy remains significantly smaller than during typical strength training (with heavy weight or high resistance), which is

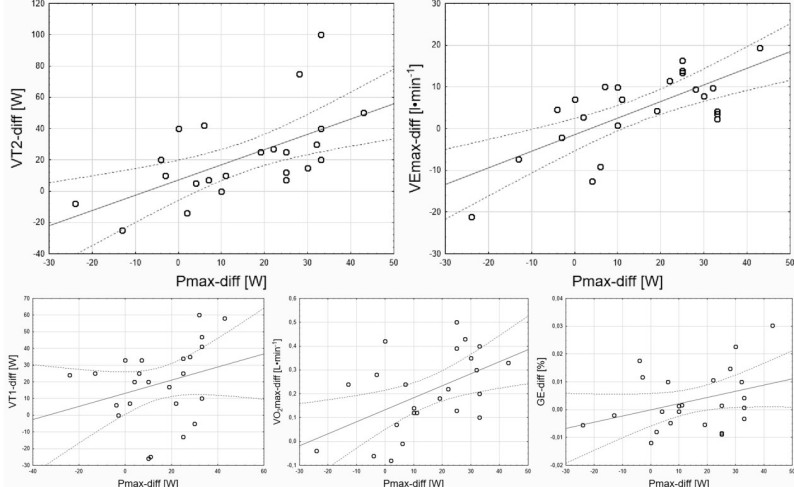

**Fig 2. Graphs illustrating the correlation between the Pmax-diff and VT2-diff, VEmax-diff, VT1-diff, VO2max-diff and GE-diff.** Pmax-diff–difference between maximal aerobic power measured after and before the POL intervention; VT2-diff–difference between power at the second ventilatory threshold measured after and before the POL intervention; VEmax-diff–difference between maximal minute pulmonary ventilation measured after and before the POL intervention; VT1-diff–difference between power at the first ventilatory threshold measured after and before the POL intervention; VO2max-diff–difference between maximal oxygen uptake measured after and before the POL intervention; GE-diff–difference between gross efficiency measured after and before the POL intervention.

important in endurance sports. Increased activation of motor units may result in a greater disturbance of muscle metabolic balance and, consequently, may stimulate improvement in physical capacity [41,42]. In order to verify this thesis among cyclists, future studies should be conducted comparing disturbances of muscle metabolic balance after intense training efforts performed at low, high and freely chosen cadence among women.

Based on the analyses performed in the presented study, it was shown that for the entire study group of cyclists, the POL, consisting of SIT, HIIT, and LIT, improved aerobic capacity in well-trained female cyclists. Important measures of aerobic capacity, such as $VO_2max$, and Pmax, increased by 6.8% and 5.6%, respectively, in the entire study group of cyclists. However, this is a smaller change compared to the effects described in our previous studies, in which POL (which also included SIT, HIIT, and LIT training) was performed by well-trained male cyclists who achieved a 14% increase in $VO_2max$ [8,43]. In the presented study, after post-hoc analyses for the $POL_{FC}$ and $POL_{LC}$ groups, it was shown that $VO_2max$ increased by 4.6% and 8.7% while Pmax increased by 3.0% and 8.1% in the $POL_{FC}$ and $POL_{LC}$ groups, respectively. Based on the comparison between the results of men and women reported in our previous and presented study, it can be concluded that the POL may be more effective in improving aerobic capacity in trained men than in trained women. Different observations were obtained in a study conducted among untrained individuals, the results of a meta-analysis performed by Lock et al. [44] indicated that untrained men and women achieved similar improvements in Pmax, $VO_2max$ and VT2 as a result of SIT and HIIT training. Similarly, Howden et al. [45] showed that untrained men and women achieved similar improvements in $VO_2max$ after 3 months of training. However, continuing the study, the authors did not observe further improvement in $VO_2max$ in women as a result of subsequent training [45]. At the same time, the continuation of the training process among men resulted in further improvement in $VO_2max$ [45]. Helgerud et al. [46] showed that among well-trained women, the use of HIIT training resulted in a significantly greater improvement in $VO_2max$ than the use of SIT

training. However, among well-trained men, the improvement in $VO_2$max after HIIT and SIT training was at a similar significant level [46]. The results obtained by Helgerud et al. [46] may explain the observed differences in improvements of aerobic capacity between our previous studies [8,43] and the present study, because an important component of the POL intervention was SIT training. Adaptive responses among women may be suppressed because women have a higher content of slow-twitch muscle fibers compared to men [45,46]. A higher content of slow-twitch muscle fibers causes women to achieve a lower level of peripheral fatigue during repeated sprints, lower concentrations of metabolites and, consequently, lower power reduction in subsequent sprints [47]. During interval exercise, one of the metabolites formed is AMP [48,49]. AMP has an important function in exercise adaptation, as it affects the processes of angiogenesis and mitochondrial biogenesis in muscle by stimulating the kinase (AMPK) [50]. Therefore, it is possible that the lower production of AMP during sprints performed by women may be responsible for less adaptive changes to the POL, compared to studies of men.

The presented study showed that VEmax-diff and VT2-diff are factors determining Pmax-diff in the process of a polarized training program performed by female cyclists. In our previous study, it was observed that the level of VEmax measured before a training intervention could be predictive of changes in $VO_2$max resulting from the use of a polarized training program [5]. Therefore, individuals with high exercise pulmonary minute ventilation may achieve greater improvement in aerobic capacity as a result of using a polarized training program than individuals with lower pulmonary minute ventilation. During SIT and HIIT training, internal homeostasis is disturbed, which manifests in a disturbed acid-base balance. An effective training process involves systematically disrupting internal homeostasis and restoring homeostasis through rest [51]. Restoring homeostasis, after previously disturbing the acid-base balance, is facilitated by an efficient buffering process and then exhaling carbon dioxide [52]. The volume of exhaled carbon dioxide is related to the level of minute pulmonary ventilation [53]. Therefore, a high value of exercise minute pulmonary ventilation may affect on the rate of recovery during interval training and, consequently, on the power achieved in subsequent repetitions (exercises). Finally, in the presented study, the training performed in the $POL_{LC}$ group could contribute to a strong disturbance of internal homeostasis during maximal- and high-intensity exercise repetitions at a low cadence. Simultaneously, cyclists whose VEmax improved could recover faster during active rest between exercise repetitions. Thus, a polarized training program could have been more effective in improving aerobic capacity in the $POL_{LC}$ group compared to the $POL_{FC}$ group.

## Limitations

A limitation of this study is certainly the small group of participants. For this reason, the results presented in this manuscript are not representative for the general group of training female cyclists. However, it is difficult to gather a large group of professional female cyclists. Therefore, it would be very beneficial if the results of the presented study were confirmed by other researchers involved in scientific research evaluating the training process of female cyclists.

## Conclusions

A polarized training program performed by a group of well-trained female cyclists improved aerobic capacity. Performing SIT and HIIT training with a low cadence resulted in greater improvements in aerobic capacity ($VO_2$max, Pmax, VT2) in a group of well-trained female cyclists than performing these training with a freely chosen cadence.

The findings of the presented study indicate practical application for athletes and coaches in cycling. In the training process, they should take into account not only the intensity and

duration but also the cadence used during various interval training sessions, as this may have an additional impact on improving aerobic capacity.

## Author Contributions

**Conceptualization:** Rafal Hebisz, Paulina Hebisz.

**Data curation:** Rafal Hebisz, Paulina Hebisz.

**Formal analysis:** Rafal Hebisz.

**Investigation:** Paulina Hebisz.

**Methodology:** Rafal Hebisz, Paulina Hebisz.

**Project administration:** Rafal Hebisz.

**Software:** Rafal Hebisz.

**Supervision:** Paulina Hebisz.

**Validation:** Rafal Hebisz.

**Writing – original draft:** Rafal Hebisz, Paulina Hebisz.

**Writing – review & editing:** Rafal Hebisz, Paulina Hebisz.

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
