## [Decision Letter · Decision Letter 0]

26 Jul 2024

PONE-D-24-20867Greater improvement in aerobic capacity after a polarized training program including cycling interval training at low cadence (50-70 RPM) than freely chosen cadence (above 80 RPM)PLOS ONE

Dear Dr. Hebisz,

Thank you for submitting your manuscript to PLOS ONE. After careful consideration, we feel that it has merit but does not fully meet PLOS ONE’s publication criteria as it currently stands. Therefore, we invite you to submit a revised version of the manuscript that addresses the points raised during the review process.

We look forward to receiving your revised manuscript.

Kind regards,

Domingo Jesús Ramos-Campo, Ph.D

Academic Editor

PLOS ONE

Additional Editor Comments :

Please, review the article according to the reviewers comments

Reviewers' comments:

Reviewer's Responses to Questions

**Comments to the Author**

1. Is the manuscript technically sound, and do the data support the conclusions?

Reviewer #1: Yes

Reviewer #2: Partly

2. Has the statistical analysis been performed appropriately and rigorously? 

Reviewer #1: Yes

Reviewer #2: No

3. Have the authors made all data underlying the findings in their manuscript fully available?

Reviewer #1: Yes

Reviewer #2: Yes

4. Is the manuscript presented in an intelligible fashion and written in standard English?

Reviewer #1: Yes

Reviewer #2: No

5. Review Comments to the Author

Reviewer #1: Dear Authors,

I would like to express my gratitude for the opportunity to review this manuscript and congratulate for their efforts in completing this scientific work.

Monitoring training in women is a demanding process due to the specificity of hormonal changes, affecting the progress of physical performance to a greater or lesser extent. The authors carried out the training intervention reliably and in a very thoughtful approach by constructing an 8-week training with SIT, HIIT and low-intensity endurance training (LIT) in well-trained female cyclists. In order to verify the impact of this training performed at high (above 80RPM) and low cadence (50-70RPM), they not only performed a incremental test, but also conducted a verification of this test. The results of the study are clearly presented and the statistics of the study are described in detail.

However, I have a few concerns and suggestions for Authors:

In lines 50-53 referring to the study by Hebisz et all. detail the intensity of the efforts in the respective SIT, HIIT, LIT protocols and give the duration of the LIT effort.

Did you measure the participants' body mass after the training process? In Your results give the VO2max value in the unit ml/kg/min. In the POLFC group, statistical significance was observed in VO2max (l∙min-1)with non-significance relative to body mass (ml∙min-1∙kg-1), suggesting that changes in body mass may have occurred and that body mass components should be verified in future researches.

In line 278 correct the unit VO2max (correctly ml∙min-1∙kg-1 and l∙min-1).

The indicated errors do not affect the essence of this work, which can provide very important guidance in the training process in well-trained female cyclists.

I look forward to hearing from You.

Best regards,

Reviewer.

Reviewer #2: Greater improvement in aerobic capacity after a polarized training program including cycling interval training at low cadence (50-70 RPM) than freely chosen cadence (above 80 RPM)

General Comments:

The manuscript presents an investigation into the effects of two polarized training programs (POL) on aerobic capacity in well-trained female cyclists. The study is well-structured and addresses an important aspect of sports science, particularly in optimizing training methodologies for athletes. However, there are several major and minor weaknesses that need to be addressed.

Major Weaknesses:

1.Sample Size and Generalizability:

•The sample size (n=12 for each group) is relatively small, which may limit the generalizability of the findings. Larger sample sizes would provide more robust results and enhance the external validity of the study.

2.Lack of Detailed Methodological Description:

•The methodology section lacks detailed descriptions of the training protocols, including the duration and intensity of each training session. This information is crucial for reproducibility and for other researchers to apply similar protocols in their studies.

3.Statistical Analysis:

•While the manuscript reports the use of analysis of variance (ANOVA), it does not provide sufficient detail on the assumptions of the ANOVA being met (e.g., normality, homogeneity of variances). Additionally, the post-hoc tests used are not mentioned, which is essential to understand the pairwise comparisons.

Minor Weaknesses:

1.Writing Quality:

•There are several grammatical errors and awkward phrasings throughout the manuscript that need to be addressed to improve readability and clarity.

2.Literature Review:

•The introduction could benefit from a more comprehensive review of the existing literature on polarized training and its effects on different populations, not just cyclists. This would provide a broader context for the study.

3.Ethical Considerations:

•While the ethical approval is mentioned, there is no discussion on how the participants' well-being was monitored during the training sessions. This is important to ensure the safety and ethical treatment of participants.

Specific Comments:

1.Title and Abstract:

•The title is clear and informative. However, the abstract should mention the specific duration and intensity of the training programs to provide a complete overview (Lines 16-34).

2.Introduction:

•The introduction provides a good rationale for the study but lacks a detailed review of previous studies on polarized training in different populations. Including this would strengthen the background (Lines 39-40).

3.Methods:

•The description of the training protocols (Lines 18-22) is too brief. Include details on the frequency, duration, and intensity of the SIT, HIIT, and LIT sessions.

•Specify the criteria used to select the well-trained female cyclists (Lines 17-18).

4.Results:

•The statistical analysis section should mention the specific post-hoc tests used for pairwise comparisons (Lines 26-31).

•Provide a table summarizing the main results for ease of interpretation.

5.Discussion:

•The discussion should address the potential limitations of the small sample size and how it might affect the generalizability of the findings (Lines 32-34).

•Include a comparison with other studies on polarized training to contextualize the findings.

6.Conclusion:

•The conclusion is concise but should reiterate the practical implications of the findings for training programs in cyclists (Lines 32-34).

6. PLOS authors have the option to publish the peer review history of their article (what does this mean?). If published, this will include your full peer review and any attached files.

Reviewer #1: No

Reviewer #2: **Yes: **Wissem Dhahbi

---

## [Author Response · Author response to Decision Letter 0]

22 Aug 2024

Reviewer 1

Dear Authors,

I would like to express my gratitude for the opportunity to review this manuscript and congratulate for their efforts in completing this scientific work.

Monitoring training in women is a demanding process due to the specificity of hormonal changes, affecting the progress of physical performance to a greater or lesser extent. The authors carried out the training intervention reliably and in a very thoughtful approach by constructing an 8-week training with SIT, HIIT and low-intensity endurance training (LIT) in well-trained female cyclists. In order to verify the impact of this training performed at high (above 80RPM) and low cadence (50-70RPM), they not only performed a incremental test, but also conducted a verification of this test. The results of the study are clearly presented and the statistics of the study are described in detail.

- Thank you very much for your review and the valuable comments.

However, I have a few concerns and suggestions for Authors:

In lines 50-53 referring to the study by Hebisz et all. detail the intensity of the efforts in the respective SIT, HIIT, LIT protocols and give the duration of the LIT effort

- As suggested, the description of SIT, HIIT and LIT training protocols has been improved to indicate the intensity and duration of efforts.

Did you measure the participants' body mass after the training process? In Your results give the VO2max value in the unit ml/kg/min. In the POLFC group, statistical significance was observed in VO2max (l∙min-1)with non-significance relative to body mass (ml∙min-1∙kg-1), suggesting that changes in body mass may have occurred and that body mass components should be verified in future researches.

- In Table 2, we added data on participants' body mass before and after the experiment. Body mass in each of the measurements was characterized by a similarity to the normal distribution and homogeneity of variance. Therefore, we used ANOVA with repeated measures in the data analysis. We did not find an effect of repeated measures or an effect of repeated measures x group in the case of participants' body mass.

In line 278 correct the unit VO2max (correctly ml∙min-1∙kg-1 and l∙min-1).

- Thank you for catching this error, it has been corrected.

The indicated errors do not affect the essence of this work, which can provide very important guidance in the training process in well-trained female cyclists.

Reviewer 2

 Greater improvement in aerobic capacity after a polarized training program including cycling interval training at low cadence (50-70 RPM) than freely chosen cadence (above 80 RPM)

General Comments:

The manuscript presents an investigation into the effects of two polarized training programs (POL) on aerobic capacity in well-trained female cyclists. The study is well-structured and addresses an important aspect of sports science, particularly in optimizing training methodologies for athletes. However, there are several major and minor weaknesses that need to be addressed.

- Thank you very much for your review and the valuable comments.

Major Weaknesses:

1.Sample Size and Generalizability:

•The sample size (n=12 for each group) is relatively small, which may limit the generalizability of the findings. Larger sample sizes would provide more robust results and enhance the external validity of the study.

- At the end of the Discussion section, we have added a paragraph on the methodological limitations of the presented study. The problem of too small group sizes was described there. Nevertheless, we ask the Reviewer to appreciate the fact that the study on polarized training was conducted in a group of 24 female cyclists. We were not able to gather a larger group of women training in cycling. In recent years, women's sports have been reaching an increasingly higher level, which is why we wanted to conduct the study among women.

2.Lack of Detailed Methodological Description:

•The methodology section lacks detailed descriptions of the training protocols, including the duration and intensity of each training session. This information is crucial for reproducibility and for other researchers to apply similar protocols in their studies.

- As suggested, we have improved the description of training protocols by adding more details. We hope that this change will be sufficient.

3.Statistical Analysis:

•While the manuscript reports the use of analysis of variance (ANOVA), it does not provide sufficient detail on the assumptions of the ANOVA being met (e.g., normality, homogeneity of variances). Additionally, the post-hoc tests used are not mentioned, which is essential to understand the pairwise comparisons.

- The Statistical analysis section included information about the Shapiro-Wilk test. It was also written that in the case of RER-rec, the requirement of similarity to the normal distribution was not met. However, we did not perform an analysis of variance homogeneity. This has been corrected in the current version of the manuscript. Levene's test showed that VT2 did not meet the requirement of variance homogeneity. Therefore, we used the Friedman nonparametric test to analyze RER-rec and VT2.

In the case of ANOVA analysis, the Scheffe test was used as a post-hoc test.

In Table 2, the statistical probability level p for RER-rec was incorrectly marked. This error has been corrected.

Minor Weaknesses:

1.Writing Quality:

•There are several grammatical errors and awkward phrasings throughout the manuscript that need to be addressed to improve readability and clarity.

- The text of the manuscript was checked and some sentences were corrected by a translator with whom our University cooperates.

2.Literature Review:

•The introduction could benefit from a more comprehensive review of the existing literature on polarized training and its effects on different populations, not just cyclists. This would provide a broader context for the study.

- In the Introduction section, a paragraph was added regarding the impact of a polarized training program on the physical capacity and performance of athletes in other sports disciplines.

3.Ethical Considerations:

•While the ethical approval is mentioned, there is no discussion on how the participants' well-being was monitored during the training sessions. This is important to ensure the safety and ethical treatment of participants.

- In the Materials and Methods section, we added information that each participant had to have consent to participate in training from a sports medicine physician. In addition, each participant could withdraw from the experiment at any time without giving a reason. In the event of observing any ailments (e.g. hypertension, tachycardia, dizziness, shortness of breath) – participants were to withdraw from further participation in the experiment. 

- In addition, to prevent overreaching and ensure the well-being of the participants, recovery microcycles were used in which the cyclists reduced the volume of the main part of the training session by 50%, similar to the assumptions of undulating periodization of training (Holmes et al. 2018). This is shown in the new figure (Figure 1). We did not include this information in the first version of the manuscript because we thought it would add confusion to the description of the training intervention.

Specific Comments:

1.Title and Abstract:

•The title is clear and informative. However, the abstract should mention the specific duration and intensity of the training programs to provide a complete overview (Lines 16-34).

- As suggested, additional information has been added.

2.Introduction:

•The introduction provides a good rationale for the study but lacks a detailed review of previous studies on polarized training in different populations. Including this would strengthen the background (Lines 39-40).

- In the Introduction section, a paragraph was added regarding the impact of a polarized training program on the physical capacity and performance in different populations

3.Methods:

•The description of the training protocols (Lines 18-22) is too brief. Include details on the frequency, duration, and intensity of the SIT, HIIT, and LIT sessions.

- As suggested, information about the frequency, intensity and duration of SIT, HIIT and LIT sessions has been added.

•Specify the criteria used to select the well-trained female cyclists (Lines 17-18).

- As suggested, the criteria were specified: "The participants of the experiment were considered as well-trained based on their baseline VO2max and training experience, this classification was presented by Decroix et al. (2016)."

They are mentioned in the Abstract and in the Materials and Methods section.

4.Results:

•The statistical analysis section should mention the specific post-hoc tests used for pairwise comparisons (Lines 26-31). 

- This has been corrected by mentioning specific tests: "Post-hoc Scheffe analysis"

•Provide a table summarizing the main results for ease of interpretation.

- We are very sorry, but we do not understand this comment. The main results are in table 2 and if we create a new table we will duplicate these results.

5.Discussion:

•The discussion should address the potential limitations of the small sample size and how it might affect the generalizability of the findings (Lines 32-34).

- As suggested, we addressed the limitation related to the small sample size in the final part of the discussion.

6.Conclusion:

•The conclusion is concise but should reiterate the practical implications of the findings for training programs in cyclists (Lines 32-34).

- As suggested, we have added a practical implication in the Conclusion section of the manuscript body: "The findings of the presented study indicate practical application for athletes and coaches in cycling. In the training process, they should take into account not only the intensity and duration but also the cadence used during various interval training sessions, as this may have an additional impact on improving aerobic capacity."

And a shorter version in the Abstract: "This finding is a practical application for athletes and coaches in cycling, to consider not only the intensity and duration but also the cadence used during various interval training sessions."

---

## [Decision Letter · Decision Letter 1]

25 Sep 2024

Greater improvement in aerobic capacity after a polarized training program including cycling interval training at low cadence (50-70 RPM) than freely chosen cadence (above 80 RPM)

PONE-D-24-20867R1

Dear Dr.HebiszWe are pleased to inform you that your manuscript has been judged scientifically suitable for publication and will be formally accepted for publication once it meets all outstanding technical requirements.

Within one week, you will receive an e-mail detailing the required amendments. When these have been addressed, you will receive a formal acceptance letter and your manuscript will be scheduled for publication.

If your institution or institutions have a press office, please notify them about your upcoming paper to help maximize its impact. If they will be preparing press materials, please inform our press team as soon as possible -- no later than 48 hours after receiving the formal acceptance. Your manuscript will remain under strict press embargo until 2 pm Eastern Time on the date of publication. For more information, please contact onepress@plos.org.

Kind regards,

Domingo Jesús Ramos-Campo, Ph.D

Academic Editor

PLOS ONE

Additional Editor Comments (optional):

Reviewers' comments:

Reviewer's Responses to Questions

**Comments to the Author**

1. If the authors have adequately addressed your comments raised in a previous round of review and you feel that this manuscript is now acceptable for publication, you may indicate that here to bypass the “Comments to the Author” section, enter your conflict of interest statement in the “Confidential to Editor” section, and submit your "Accept" recommendation.

Reviewer #1: All comments have been addressed

2. Is the manuscript technically sound, and do the data support the conclusions?

Reviewer #1: Yes

3. Has the statistical analysis been performed appropriately and rigorously? 

Reviewer #1: Yes

4. Have the authors made all data underlying the findings in their manuscript fully available?

Reviewer #1: Yes

5. Is the manuscript presented in an intelligible fashion and written in standard English?

Reviewer #1: Yes

6. Review Comments to the Author

Reviewer #1: Dear Authors,

Thank you for considering my suggestions and incorporating them into the new version of the manuscript. The document quality has been improving throughout the review process. I accept the submitted manuscript.

Best regards,

Reviewer

7. PLOS authors have the option to publish the peer review history of their article (what does this mean?). If published, this will include your full peer review and any attached files.

Reviewer #1: No

---

## [Editor Report · Acceptance letter]

4 Nov 2024

PONE-D-24-20867R1 

PLOS ONE

Dear Dr. Hebisz, 

I'm pleased to inform you that your manuscript has been deemed suitable for publication in PLOS ONE. Congratulations! Your manuscript is now being handed over to our production team.

Kind regards, 

on behalf of

Dr. Domingo Jesús Ramos-Campo 

Academic Editor

PLOS ONE